# The Need to Articulate Historic and Cultural Dimensions of Landscapes in Sustainable Environmental Planning—A Swedish Case Study

**Ingegärd Eliasson [1,\*], Susanne Fredholm [1], Igor Knez [2] and Eva Gustavsson [1]**

1    Department of Conservation, University of Gothenburg, 405 30 Gothenburg, Sweden
2    Department of Occupational Health Science and Psychology, University of Gävle, 801 76 Gävle, Sweden
\*    Correspondence: ingegard.eliasson@conservation.gu.se; Tel.: +46-317862832

**Abstract:** Ignoring the historic and cultural dimensions of landscapes makes environmental planning unsustainable, which in the long run, will have a negative impact on both the environment and society. This paper examines the work and perceptions of practitioners with a focus on the role of historic and cultural landscape dimensions and their relation to the recent implementation of the ecosystem service framework in sustainable environmental planning. Semi-structured interviews with officials at local and regional planning levels in a Swedish case study showed that the historical landscape forms the basis for environmental work. Respondents expressed an integrated view of the landscape, and historic and cultural landscape dimensions were considered important in the initial planning process. However, several challenges existed later in the planning process and final decision-making, such as conceptual ambiguities, unclear policy and assignments, limited cross-sectorial coordination and lack of awareness, knowledge, resources and other priorities. The results also show that the respondents worked regularly with intangible landscape dimensions, which can be defined as cultural ecosystem services, but they do not label them as such. Furthermore, established knowledge and expertise of heritage planning was not activated in the implementation of the ecosystem service approach. We conclude that historic and cultural landscape dimensions are not ignored in practice, but there is a need to articulate these aspects more clearly in order to achieve sustainable environmental planning. There is also an unexplored opportunity to connect skills and create new forms of cross-sectorial collaboration between heritage planning and the ES approach.

**Keywords:** cultural heritage; cultural ecosystem services; historic environment; spatial planning; landscape policy; sustainable development

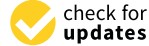



## 1. Introduction

To achieve sustainable environmental planning, the whole environment must be considered, but previous research shows that historical and cultural perspectives on the landscape are often ignored [1–9] (The historic dimensions, "the surviving physical impacts of people on the landscape" and the cultural dimensions, "the intangible meanings, values, attributes and associations that people attach to its physical components" of landscapes [10] are integrated parts of the highly dynamic concept of heritage [11,12].

Guided by international charters and conventions, heritage conservation principles have moved from a focus on preserving historic monuments and sites towards a more integrative and people-centred focus on using the past in the present [13] and to manage change sustainably. This future-oriented approach includes not only the care for landscape materiality and characteristics but also local knowledge and history, stories and myths, crafts, minority and local identity, and collective memory [7,14–17]. The application of contemporary heritage conservation principles in local and regional planning is today a well-established practice in many parts of the world [17–21] with a vast variety of professional profiles, approaches and agendas.

With the adaptation to the principles of sustainable development [22] heritage has become more relevant in an environmental planning context. In 2005, the Council of Europe (COE) adopted the Faro Convention, emphasizing the role of heritage as a resource for sustainable development in society [23]. In 2017, the COE adopted the European Cultural Heritage Strategy for the 21st century, stressing that heritage cannot be viewed in isolation from its physical and cultural context, and that developing a high-quality living environment means considering heritage in other sectoral policies, such as development, environmental conservation and land use planning [24]. Sustainable heritage is today a developing field guided by the urgent need to address social-cultural and environmental issues together [9]. However, as argued by [25] the conservation and sustainability fields has not yet recognized nor elaborated "the full implications of considering the heritage dimension and value of nature in unison".

Integrative instruments for environmental planning are requested by both scientists and practitioners. One such instrument is the Ecosystem Service (ES) framework developed to assist environmental decision-making. ES aims to bridge natural and social sciences and facilitate collaborative management in a shared framework for assessing values in the landscape [6,26–30]. In the ES framework, the notion of landscape and its historic and cultural dimensions is central to debates on loss of biodiversity, and mostly relevant in relation to cultural ecosystem services (CES). Six categories of CES have been defined [26] including: cultural identity (the current cultural linkage between humans and their environment); heritage values ("memories" in the landscape from past cultural ties); spiritual services (sacred, religious, or other forms of spiritual inspiration derived from ecosystems); inspiration (the use of natural motives or artefacts in arts, folklore, and so on); aesthetic appreciation of natural and cultivated landscapes; and recreation and tourism. While the MA classification of ES has been developed and refined [28,30,31] heritage remains a relevant concept, as the ES approach integrates historic and cultural perspectives on landscapes and biocultural heritage into decision-making processes. However, there is a shortage of studies focusing on cultural heritage and identity and only a handful of existing ES publications gives a more comprehensive description of "newer notions of heritage" in relation to landscape [32] Another shortcoming is the focus on immaterial aspects through heritage values, which excludes the material aspects of human relations with the environment [25].

In the recent decade, due to a directive by the European Commission the EU member states have started to implement the ES approach in practice. However, the intangibility of CES, evaluation difficulties, methodological and conceptual issues as well as the fundamentally instrumental framing of the ES framework are reported to restrict the integration of CES in environmental planning and policies [6,25,32–39]. A closer collaboration and exchange of knowledge between heritage planning and the ES approach can be of mutual benefit as the methodological framework of ES, where the cultural benefits of nature are explored as resources for society, is in line with current approaches to heritage planning [16,21,32,40–42]. Taking advantage of established discourses, approaches and practices in heritage planning could possibly improve the understanding of CES beyond the current focus on recreation. Heritage planning would also benefit from such a connection as the ES approach provides a methodological framework to bridge the gap between heritage and sustainable development and to recognize cultural landscapes and natural features with cultural significance (natural heritage) in environmental planning [21,26,32,40–42].

As shown above, the historic and cultural perspectives on landscape often have little significance in sustainable environmental planning. This also applies to the implementation of ES, despite the framework having been developed with the purpose to bridge natural and social science, and facilitate collaborative management and communication across sectors. Ignoring the historic and cultural dimensions of landscapes makes environmental planning unsustainable, which in the long run will have a negative impact on both the environment and society. In theory, a closer collaboration between the fields of heritage planning and the ES can be of mutual benefit as suggested by previous studies. However, few studies

have examined how practitioners understand and approach the role of historic and cultural landscape dimensions for sustainable development in environmental planning. In order to fill this gap, the aim of the present study was to examine the work and perceptions of practitioners with a focus on discussing how historic and cultural landscape dimensions are understood and articulated and what opportunities exist for intersectoral exchange of knowledge, in light of the recent implementation of the ES approach. For this purpose, we used data from a case study of sustainable environmental planning at local and regional public agencies in Sweden. The study was guided by the following research questions:

- What are the opportunities and challenges to ensure consideration of the historic and cultural dimensions of landscapes?
- Which historic and cultural landscape dimensions are considered?
- What is the awareness and knowledge about the concepts of ES and CES?
- Is the implementation of ES and understanding of CES related to the established practice of heritage planning?

## 2. Materials and Methods

This paper is part of an interdisciplinary research project with the aim to investigate the role of cultural heritage and the historic environment in sustainable landscape management [43,44]. The present paper draws upon data collected from interviews. A qualitative, single case study approach [45–48] was used to gain a deeper understanding of the issues at hand. The interdisciplinary project team included researchers with expertise in the field of physical geography, conservation of the built environment, psychology and biocultural heritage.

### 2.1. The Case Study Area

The case study area, the Lake Vänern Archipelago Biosphere Reserve, includes the municipalities of Mariestad, Götene and Lidköping with a total population of 80,000 inhabitants (Figure 1). The geographical area includes parts of Lake Vänern, the largest lake within the European Union, and an arable landscape with a varied topography consisting of post-glacial clay plains, mylonite intrusions, glacial moraine deposits and a Cambro-Silurian flat-topped mountain named Kinnekulle. People have lived in the area for at least 6000 years, and a richness in landmarks and artefacts dating back to the Bronze Age imply millennia of cultivation and influence on the landscape which is still visible in the diversity of plant species [49]. Biosphere reserves are intended as model areas for sustainable development and, in order to be designated by UNESCO's Man and the Biosphere Programme (MAB), the landscape must include both a rich cultural heritage and high levels of biodiversity [50]. The global strategy for MAB aims to guide the local regional and national implementation of Agenda 2030 by integrated planning and landscape management [51].

The Swedish Environmental Code together with the Planning and Building Act form the legal basis for environmental planning, including heritage planning, in Sweden. The aim of the Environmental Code [52] is to promote sustainable development, which will assure a healthy and sound environment for present and future generations. "The environment" is used in broad terms and includes the cultural environment. Thus, the regulatory framework makes no clear distinction between natural and cultural values of landscapes, and the Environmental Code shall be applied in such a way as to ensure that valuable natural and cultural environments are protected and preserved in combination. Furthermore, the cultural environment serves as an important aspect in sustainability policy. For example, the Swedish National Heritage Board, under the Ministry of Culture, is in charge of the 2030 Vision for cultural heritage management in Sweden. An important goal of the vision is to increase awareness that cultural heritage and the cultural environment are important parts of the work for a sustainable inclusive society [53]. Additionally, the current Swedish environmental policy includes a "generational goal," which is intended to guide environmental action at every level of society by means of 16 environmental

quality objectives and a number of milestone targets. One important target, decided by the Swedish parliament, is that a majority of the municipalities, by 2025, shall integrate ecosystem services in planning, building and management of the urban built environment. To accomplish this target and the environmental policy in practice the municipalities use the detailed development plans which are legally binding and the most important instrument. The municipalities have the main responsibility for environmental planning in Sweden in dialogue with the regional planning level, i.e., the County Administrative Boards. Implementation of the ES framework has gradually increased at the regional and local planning levels in Sweden during the past 10 years, as a consequence of governmental decisions and development of national policies. Previous studies report a limited awareness of the concept of ES and a slow integration at local and regional planning levels [21,38,54] but, during recent years, the integration has increased as shown by [55]. Still, several barriers exist, such as the use of different definitions, approaches and methods as well as a lack of bridging perspectives and traditional division between nature and culture [21,38,56,57]. A recent study [55] report a "heterogeneity in the degree, focus and strategy" among the municipalities and a "gap between visions, strategic goals and their implementation" from a review of 231 Swedish municipal comprehensive plans.

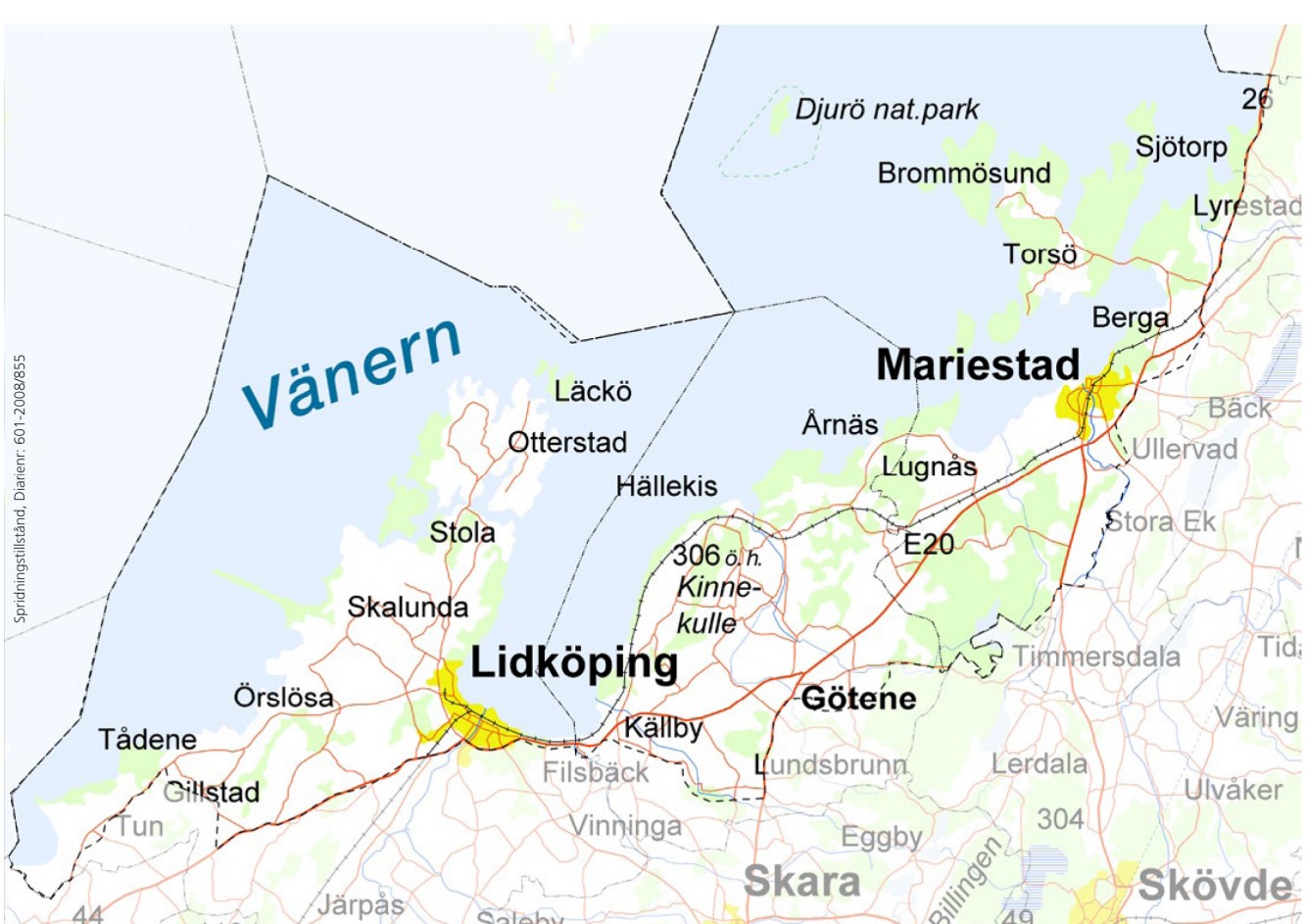

**Figure 1.** Lake Vänern Archipelago Biosphere Reserve in Sweden includes the three municipalities, Mariestad, Götene and Lidköping. Source: Lantmäteriet state of dispersal Dnr 601-2008-855.

*2.2. Interviews*

In total, eighteen semi-structured interviews were conducted between 2018 and 2019 with officials employed or otherwise engaged by local and regional actors, including municipalities, the County Administrative Board, local museums, and the Biosphere Association. The respondents each had formal responsibilities related to the planning and management

of historic and cultural landscape dimensions but represented a range of different occupations and disciplines, such as historians, built heritage conservation, urban and landscape planning, environment conservation, engineering, and architecture. Respondents were identified through discussions in the project group and via webpages. The respondents were approached by telephone or e-mail, and they were able to choose the location for a face-to-face interview which lasted one hour on average and was recorded and transcribed. The themes and questions analyzed in the present paper are shown in Table 1.

Each interview started with a discussion about the research project and the respondents were then asked to specify their educational and professional backgrounds, as well as current work assignments. Note that no definitions of concepts related to historic and cultural landscape dimensions were given in advance, thus the respondents' answers are based on their own interpretations. Each respondent's answers were analyzed via transcripts and audio recordings. The responses were compiled in a spreadsheet and were thematically categorized based on their semantic content. Data from the semi-structured interviews are reported using "bottom up" (based on open questions) and "top down" analyses, where "the analytic process involves a progression from description, where the data have simply been organized to show patterns in semantic content, and summarized, to interpretation, where there is an attempt to theorize the significance of the patterns and their broader meanings and implications often in relation to previous literature" [46]. These analyses were done with an emphasis on listening, meaning that the interviewer actively avoided influencing and biasing the conversation [58]. In order to thoroughly understand the data, the transcripts were read several times and when needed the original audio recordings were reviewed [46].

The questions used in the semi-structured interviews were divided into 5 themes (Table 1). In two of the themes, questions appeared both in open-ended and closed-ended formats (see Table 1). The interviewer began with the open-ended question, asking the respondents to develop and discuss their answers. Next, the respondents were asked to mark multiple choices on an A4 sheet for each question. The closed-ended part of the questions also included an opportunity for the respondents to add aspects that they thought were important. When the respondent had answered the questions, the interviewer and the respondent discussed the answers and the question. Data analysis for these two themes included data both based on the answers from the audio recording and transcripts, and from the A4 sheets. The data from the A4 sheets were transferred to an Excel sheet. Percentages of total number of respondents who considered each of the aspects presented on the A4 sheets were calculated and results plotted on a spreadsheet chart. These results were again compared and sometimes complemented with authentic citations and data from the audio recording and transcripts. For the remaining three themes only open-ended questions were used (see Table 1). Each respondent's answers were analysed via transcripts and audio recordings. The responses were compiled in an Excel sheet and were thematically categorized based on their semantic content.

**Table 1.** Interview instrument—themes and questions that guided the study.

| Themes | Open-Ended Questions | Closed-Ended Questions |
|---|---|---|
| Landscape dimensions considered | Which historic and cultural landscape dimensions: Are regularly considered? Are not considered? Need more focus? | Respondents were asked to mark multiple choices related to which historic and cultural landscape dimensions they consider in their work. The respondents also had the opportunity to add choices. Red colour = regularly considered Green colour = need more focus. No mark/colour = not considered |

**Table 1.** *Cont.*

| Themes | Open-Ended Questions | Closed-Ended Questions |
|---|---|---|
| Status | What is the status of historic and cultural landscape dimensions in planning and management? | |
| Opportunities and challenges | What are your arguments and the limiting factors to ensure consideration of historic and cultural landscape dimensions in planning and decision-making? | |
| Ecosystem services | Are you familiar with the concept of ES and CES? Do you work with ES? How do you define CES? | |
| Methods and guiding policies | Which methods, directives, laws, policies, and documents guide you in your work? | Respondents were asked to mark multiple choices related to the question: "Which directives, laws, policies, and documents guide you in your work?" |

## 3. Results

Results are presented in line with the aim and research questions presented above.

### 3.1. Historic and Cultural Landscape Dimensions Are Important

One strong potential identified was that historic and cultural landscape dimensions are, according to a majority of the public officials, important in the early stages of the planning process and considered on a daily basis. The results from the interviews show that the officials have a long tradition of considering man's use of nature and of using the perspective that nature and culture is both integrated and interdependent in the landscape. One example is an EU fund LIFE-nature project entitled, "Kinnekulle Plateau Mountain—restoration and conservation" which was in progress between 2002–2007 with a budget of Euro 5.7 million. Even though this project was labelled as a "nature" restoration and conservation project the historic landscape was the base for the project. In the words of one of the respondents:

> "The job was to restore and recreate old overgrown pastures and meadows and to win back the old cultural landscape. We included quite a lot of land in this project, often not based on the natural values, instead based on, for example, old boundaries, village boundaries between grazing and outfield. The project was based on a combination of cultural history and natural values".

However, results also show several challenges to ensure consideration of historic and cultural landscape dimensions in later phases of the planning and decision process. These challenges were related to conceptual ambiguities, unclear policy and assignments, limited cross-sectorial coordination, lack of awareness, knowledge, resources and other priorities as described below.

### 3.2. Conceptual Ambiguities

The interviews revealed that public officials talk about and define historic and cultural landscape dimensions in various ways. There are conceptual ambiguities with a wide range of definitions. Some officials talk about designated buildings and areas of national interest, i.e., the material heritage that is strongly linked to the legal framework. Others talk about historic and cultural landscape dimensions in more general terms synonymous with a changing landscape, or as one official put it:

> "A process—it changes because historic and cultural dimensions are the impact of humans on the environment, the landscape and the buildings . . . ".

Some other respondents argue that the concept is a sensitive topic that tends to be perceived as fluctuant opinions, based on attitudes, rather than professional judgments. In order to increase the influence of historic and cultural landscape dimensions in current practices they describe how they have to find innovative ways, as for example showing its instrumental value for the tourism industry.

### 3.3. Unclear Policy and Assignments

The conceptual ambiguities are closely related to unclear policy and assignments as reported by several respondents. A partial or complete lack of political governance documents and policy, including cultural heritage management programs, is one reason. Another reason put forward is that the politicians do not always use their own governing documents. One respondent expresses the unclear policy and assignments in the following way:

> "After all, there is a political program, where you put forward nice wishes about how things should be. The ideas are put into an action plan where you make priorities, but historic and cultural values are rarely highlighted, so we lose it on the way. You continue to work and continue to feel that you have no control. I would like to have an action plan with headings, such as cultural heritage, that include the activities we have on this topic. What is the focus of our attention? How can we work successfully? At present, some idea pops up and, yes, we implement it but our work on cultural heritage becomes event driven."

Several respondents argue that the only way to increase the status of historic and cultural landscape dimensions in environmental planning is to promote political decisions that make cultural heritage programs and other policy documents an integral part of the municipal comprehensive plans. A majority of the respondents answered that they used the National environmental quality objectives, Planning and Building Act, local cultural heritage programs and the Historic Environment Act for guidance in their work. A few respondents argued that the ideas of the European Landscape Convention (ELC) inspired them in their work although less than a third used it as a guiding document on a daily basis.

### 3.4. Limited Cross-Sectorial Coordination

Even though most respondents share a view of the integrated landscape where nature and culture are interdependent, the idea that aspects related to historic and cultural landscape dimensions belong in a certain department, such as the planning department, is deeply rooted. In the words of one respondent who worked with local history and stories in networks of local stakeholders, and NGOs such as local museums and collections of industrial heritage, e.g., sawmills, stone masonry, dairies, machines, etc.:

> "I don't consider myself to work with historic and cultural landscape dimensions as I don't work with physical places . . . there has been no political awareness and much of this work seems to belong to the planning office . . . much has been left to the non-profit NGOs . . . I can't say with good conscience that we have clear structures. We would, if we pushed in the same direction and knew what we wanted, be a force in the right direction. Instead, it is a force that spreads in all possible directions . . . "

Existing sectoral funding and organization at national, regional and local planning levels are one reason for the limited cross-sectorial coordination. A majority of the respondents argued that increased cooperation between local and regional agencies and other actors is required to integrate the cultural and natural landscapes values to reach a sustainable development (Eliasson et al. unpublished).

### 3.5. Lack of Awareness, Knowledge, Resources and Other Priorities

Respondents argued that there is a lack of awareness and knowledge among politicians, developers and the public about the role of historic and cultural landscape dimensions in environmental planning. This is primarily related to individual politicians rather than political parties. The political term of office of only four years makes the change of individual politicians an educational challenge for the public officials. In Sweden, the public's influence on the planning process is ensured through the National Planning and Building Act. Respondents describe that sometimes when they talk with, for example, house-owners about historic and cultural landscape dimensions it creates tensions. Landowners often consider the Historic Environment Act as a threat to them as they think that increasing awareness of historic and cultural values will increase costs and mean limitations on their land. Developers fear increased costs and delays. Some respondents describe difficulties in making politicians, developers and the public listen to facts. Or, in the words of one respondent:

> "Ignorance is an incredible resistance, especially when it is paired with fact resistance and denial of knowledge".

Education is a solution to the lack of awareness and knowledge according to some of the respondents who were engaged in education of politicians and the public. However, the respondents did not use public participation methods on a regular basis. Most respondents argue that the organization's finances and priorities are one important constraint leading to a limited budget for heritage planning.

### 3.6. Historic and Cultural Landscape Dimensions Considered

The historic and cultural landscape dimensions most often considered by the public officials on a regular basis (Table 1, closed-ended questions) are shown in Figure 2. More than sixty percent of the public officials claim they regularly (daily) work with buildings and industrial heritage. More than fifty percent of the respondents worked regularly with place identity, agricultural environments and green areas, and 45% worked with aspects of landscape views and local history on a daily basis (Figure 2). Thus, it is evident that the public officials regularly worked with both tangible and intangible heritage. The open-ended questions revealed (Table 1) a close connection between the tangible and intangible heritage as the respondents worked with aspects such as local identity, history and shared stories in relation to the industrial, agricultural and built-up heritage.

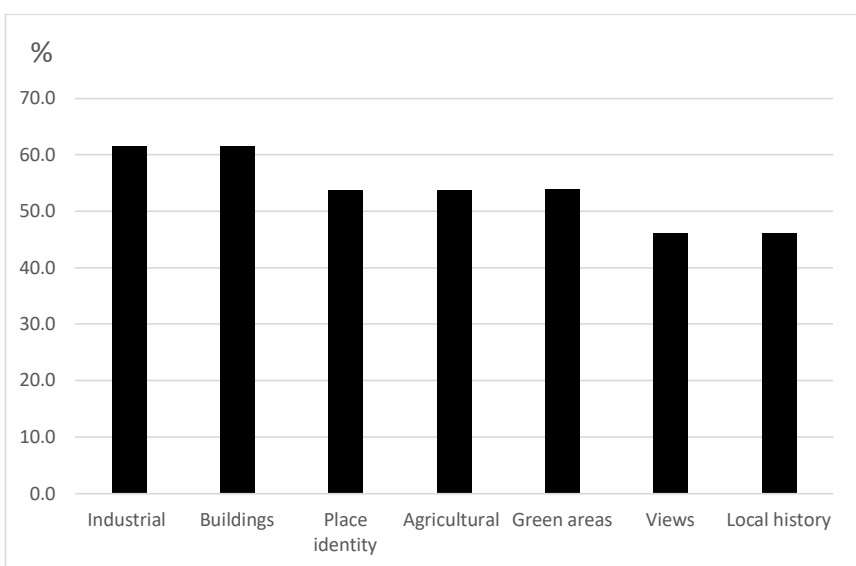

**Figure 2.** The *x*-axis shows the seven most important historic and cultural landscape dimensions that are regularly considered in the planning process (measured as percentage, *y*-axis). See also Table 1, closed-ended questions.

Respondents were also asked which historic and cultural landscape dimensions that need more focus (Table 1, closed-ended questions). As shown in Figure 3, more than fifty percent of the public officials wanted to focus more on local history. Other dimensions that need more focus in the planning process, according to a third of the respondents, are meeting places, the dark heritage [59], experience, agricultural environments, local knowledge and stories.

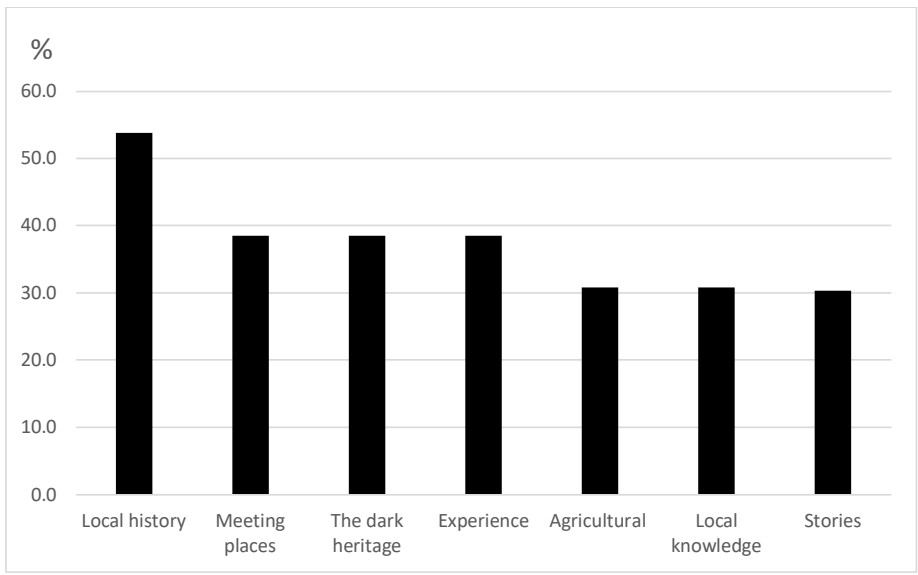

**Figure 3.** The *x*-axis shows the seven most important historic and cultural landscape dimensions that need more focus in the planning process (measured as percentage, *y*-axis). See also Table 1, closed-ended questions.

One of the public officials expressed a desire to "fill the historic environments with this intangible cultural heritage". At the same time, a majority of the respondents claim that they have limited opportunities to do so because of budget priorities, lack of methods and sometimes lack of skills.

*3.7. Awareness, Knowledge, and Implementation of the ES and CES*

All of the public officials knew about ecosystem services at the time of the interviews but only half of them worked with the ES approach. Of those who did not, several referred to the municipal ecologist and the nature conservation unit for information about ES implementation. It was clear that implementation of the ES approach had just begun. The respondents who had started to implement the ES approach responded that, at this early stage, the work was tentative and that they needed methods that can be used in the planning process. Most respondents considered the ES approach to be a benefit in the planning process but expressed a great deal of uncertainty about how to work with it and how to convince others in the organization.

The concept of cultural ecosystem services was less known. Half of the respondents had heard about CES but only a quarter had worked with it. One of the respondents who worked with CES argued that the landscape history becomes a tool for reaching out to people. The respondent used storytelling as a way to communicate the relation of the importance of the cultural-historical landscape to, for example, old wells and people's traditional use of water and herbs, including the historical monks' knowledge of the curative effect of plants. Another respondent worked with CES in a GIS-based green structure project of biotope mapping in municipal comprehensive plans. In this respect, CES was considered indirectly through sensual experiences, such as the rustling of leaves, birdsong and green leafy environments that have a direct bearing on the nature and park types that were identified. Yet, another respondent worked with CES on a GIS-based

project aimed at identifying easily accessible attractive recreational environments. Here, information was used about nature and cultural reserves, areas of national interest for cultural heritage and for outdoor life, national parks, etc. The purpose was to make people understand their context, their background and their identity and the need to preserve the structures in the landscape. Even though these three examples touch upon aspects of, or are based on, historic and cultural landscape dimensions neither of them involved established knowledge of heritage planning.

Respondents argued that increased cooperation between sectors and with the civil society is the key in the implementation of Agenda 2030 and ES. An example of a public participation initiative at municipality level to communicate ES and CES via the national environmental quality objectives was through outdoor events for the public. The basic idea is that people should experience ecosystem services directly in the landscape. One example of an environmental station event was based on the environmental quality objective, "Flourishing lakes and streams". The event was located at Lake Vänern and the municipality invited the public and offered four different dishes (450 portions) of fish (Zander also named pikeperch). A professional angler made a presentation about Zander fishing, what it looks like and what actions are implemented to make the fishing industry sustainable in Lake Vänern. Additionally, representatives from the County Administrative Board and the biosphere reserve office took part in the event. Cooperation between different actors in the biosphere reserve is the key solution, as expressed by one respondent:

> "The events are under the auspices of the municipality although the biosphere office and County Administrative Board are involved players. We invite them and they become part owners of these project ideas and arrangements. Thus, we get a greater efficiency and gain more expertise when we create the events. It is important to find a form of co-creative processes so that everyone feels that they own the products. If you have confidence in each other, it will give results."

A new initiative for cooperation between different actors within the biosphere reserve is a ES network established a few years ago by a public official at municipality level in order to increase ES cooperation between the three municipalities in the biosphere reserve. The motive was to find a structure for the ecosystem services in the historic landscape and to find a horizontal platform where you trust each other and find common targets. The ES network consists of representatives from the three municipalities, the County Administrative Board, the biosphere reserve office and two nature conservation associations. The ES network meets regularly and has applied and received funds for training projects in ecosystem services for politicians, public officials and developers working within the biosphere reserve. The ES network sometimes invites researchers and actors working with historic and cultural landscape dimensions to their meetings, but they are not regular members of the ES network.

## 4. Discussion

### 4.1. Opportunities and Challenges to Ensure Consideration of the Historic and Cultural Dimensions of Landscapes

The results show that the historic and cultural landscape dimensions are important aspects in the early stages of the planning process at local and regional levels. This is partly due to current legislation, as consideration of the cultural environment is primarily governed by the Planning and Building Act, but also by the Roads Act and the Act on Railway Construction. The Swedish environmental quality objectives also contain requirements for consideration of the cultural environment. In combination with the respondents' long tradition of considering man's use of nature with an integrated landscape perspective this gives a good potential to ensure consideration of historic and cultural landscape dimensions in environmental planning. These results are in line with a parallel study in the Swedish mountains showing that practitioners strive for an integrated management of historic and cultural landscape dimensions and that these values are considered throughout the planning process [21]. One could argue that these results are in contrast to many

previous studies showing a limited consideration of historical and cultural perspectives in environmental planning [1–9]. However, there is a clear conceptual ambiguity among respondents who define historic and cultural dimensions of landscapes quite differently, ranging from predominantly material aspects protected by law to intangible aspects, which are not as effectively managed in planning contexts. The respondents do not understand, nor talk about, landscape complexities in a consensual manner.

Furthermore, lack of methodologies to include local communities other than through formal consultation in the planning process makes the intangible and socio-culturally perspectives on landscape less dominant, which is the norm in the Swedish context [20,60]. This is in line with previous research highlighting the need for local participation in order to manage cultural landscapes [7,61–63]. Other challenges identified in the present study are unclear assignments and lack of political policy as well as unawareness and disinterest among politicians, developers and the public. The public officials expressed a wish to promote political decisions that connect historic and cultural landscape dimensions to environmental planning and especially the implementation of ES in combination with training efforts. These results are in line with other studies [55,64] that argue for increased political support and capacity building initiatives in order to facilitate ES implementation in municipal planning practice. Results showing that the CES concept is far from practically implemented in policies were also reported by [39] from a survey with experts on agricultural landscapes. The ELC, with its landscape policy focusing on sustainable development and the cultural dimensions of the landscape [65] is important in this respect. However, the results show that only a third of the public officials interviewed used ELC for guidance. This result is in line with previous studies showing a limited impact of ELC in practice [21,66,67]. Another challenge identified is that the idea that historic and cultural landscape dimensions belong to a certain department, such as the planning department, is deeply rooted among public officials. Professional roles and responsibilities are established mainly because of sectoral funding and the organization. These results confirm previous studies showing that, despite its integrative ambition, implementation of the ES concept often lack horizontal integration between sectors and is highly influenced by established endorsed professional roles and responsibilities in land-use planning [16,21,38,56,68].

### 4.2. Heritage Planning and the ES Approach

Heritage values, place identity, landscape views, local history, local knowledge, stories, and sensory experiences are examples of historic and cultural landscape dimensions that were regularly addressed in local and regional planning in the present study. These intangible landscape dimensions fit into the six categories of CES, defined by [26], but the respondents did not label them as CES. The public officials regularly worked with, and highlighted, the importance of intangible landscape dimensions and expressed a desire to work more with these values. Moreover, most public officials have a long tradition of considering man's use of nature with an integrated landscape perspective. These results are in line with [21] who show that local history, identity, stories and collective memory are aspects regularly considered by officials, in the County of Jämtland Sweden, and practitioners also strive, despite difficulties, to use a coherent landscape approach in planning. An integrated landscape approach to planning is in line with the ES approach and would facilitate a closer collaboration between actors involved in heritage planning and the ES implementation. Nevertheless, at the time of our interviews, established knowledge and expertise of heritage planning was not activated in the implementation of the ES approach. Respondents expressed that CES, are generally difficult to concretize, and thus are given less consideration than other ecosystems services. These results are in line with previous findings showing a limited consideration of cultural heritage within ES research and practice [21,25,32,41,42].

There is certainly an opportunity to connect heritage planning with the ES approach. This, however, requires a clarification and mapping of actors working with historic and cultural landscape dimensions within the regional and local agencies, NGOs and civil

society. It is especially important to identify actors who do not consider themselves as working with the "right" aspects of heritage or the "right" department/organization. The definition and role of CES in the ES approach must be developed in order to be able to connect skills and create new forms of collaboration integrating historic and cultural landscape dimensions in environmental planning. A neutral platform could support the connection of historic and cultural landscape dimensions and ES in sustainable environmental planning. Several of the respondents interviewed argue that the Lake Vänern Archipelago Biosphere Reserve could be an arena for a new understanding of the landscape, beyond the established permanent division between nature and culture in formal environmental planning. However, as shown by [69], it is only a few of the public officials at local and regional planning levels that regularly cooperate with the biosphere reserve organization. The need to "create communities of practice", with shared goals in the implementation of ES and sustainable development was suggested by [38]. The new ES network, established by public officials at municipality level in the Lake Vänern Biosphere Reserve, has successfully created new forms of collaboration and an extended dialogue about ES between various actors at different levels, including training programs for politicians and developers. Even though the ES network sometimes invites researchers and professionals working with historic and cultural landscape dimensions these actors are not regular members of the ES network. Ideally, a network for ES implementation needs to include actors with different knowledge and perspectives to meet the basic requirements of the ES approach, i.e., all four categories of ecosystem services must be considered to reach sustainable development.

Municipal heritage planning would also most probably benefit from a closer connection to the ES framework. The critique against the artificial separation between natural and cultural heritage is well established in research and practice, and in the wake of "embracing this dissolution" [70], new methods and tools for more inclusive landscape interpretations are being developed. In such system-based approaches, the connections and benefits of heritage tied to natural resources are explored [71] (as well as the role of "living heritage" in socio-ecological systems [72]. These value-driven approaches are grounded in an understanding of the cultural landscape to include both man-made and non-human-made structures, and they often require a combination of qualitative methods to enable evaluation of CES, such as stated preference methods, workshops, etc. Although public participation has been on the heritage planning agenda since the 1990s, it is still an "unfixed, uncertain and contested concept" [60], and not practiced in everyday planning situations. This is in line with [7] who argue that management strategies and conservation policies based exclusively on decision-makers criteria are counterproductive for the conservation of cultural landscapes. Participatory ES assessment methods currently being developed for local resource users to identify and evaluate the key services of a particular ecosystem, including heritage, could be a way forward.

### 4.3. Concluding Remarks

Are historic and cultural landscape dimensions ignored in environmental planning? The price of ignoring these values is an unsustainable environmental planning and according to literature historic and cultural perspectives often have little significance in environmental planning. However, only a few studies take a deep dive into practice and study the work and perceptions of practitioners. The present study contributes to increased knowledge about the role of historic and cultural landscape dimensions in sustainable environmental planning and the compatibility and commonality of the fields of heritage planning and the implementation of the ES approach. In contrast to many other studies, we can conclude that historic and cultural landscape dimensions were considered important by practitioners who in general had a holistic view of the environment where nature and cultural values of the landscape are interdependent. However, in line with previous studies, several challenges exist throughout the planning process as shown above. Our conclusion is that historic and cultural landscape dimensions are not ignored in practice, but there is a need to articulate these aspects more clearly in order to achieve sustainable environmental

planning. Established knowledge and expertise of heritage planning was not activated in the implementation of the ES approach. Interesting is, however, that the practitioners worked with intangible landscape dimensions, which can be defined as CES. Thus, there is an unexplored opportunity to connect skills and create new forms of cross-sectorial collaboration between heritage planning and the ES approach. Results and conclusions presented in this paper are in line with results from a parallel study in the Swedish mountains [21]. The agreement between these two studies strengthens the validity of the results but a validation through future interdisciplinary studies of the work and perceptions of practitioners in other parts of the world is welcomed.

**Author Contributions:** I.E.: Conceptualization, Methodology, Validation, Formal Analysis, Investigation, Writing—original draft, Supervision, Project Administration, Funding acquisition. S.F.: Conceptualization, Methodology, Formal Analysis, Investigation, Writing—Review and Editing. I.K.: Conceptualization, Methodology, Writing—Review and Editing. E.G.: Conceptualization, Writing—Review and Editing. All authors have read and agreed to the published version of the manuscript.

**Funding:** This research was funded by the Swedish National Heritage Board (Dnr 3.2.2-5202-2016) and is part of the project "Cultural heritage and the historic environment in sustainable landscape management". Co-funding was also received from University of Gothenburg, Sweden and University of Gävle, Sweden.

**Institutional Review Board Statement:** Semi-structured interviews with public officials were conducted in accordance with the ethical guidelines of the University of Gothenburg Sweden. The study was performed in accordance with the ethical standards of The European Code of Conduct for Research Integrity—ALLEA.

**Informed Consent Statement:** Informed consent was obtained from all individual respondents interviewed. The material from the interviews has been de-identified in the results presented in the paper.

**Conflicts of Interest:** The authors have no relevant financial or non-financial interests to disclose. The authors have no competing interests to declare that are relevant to the content of this article. All authors certify that they have no affiliations with or involvement in any organization or entity with any financial interest or non-financial interest in the subject matter or materials discussed in this manuscript. The authors have no financial or proprietary interests in any material discussed in this article.

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
