# Peer review of "The Need to Articulate Historic and Cultural Dimensions of Landscapes in Sustainable Environmental Planning—A Swedish Case Study"

_land, doi:10.3390/land11111915_

Round 1

Reviewer 1 Report

Thank you for sharing this draft with me. I appreciate all of the details included like tables with your research questions and methods. I would be curious to know if a follow-up study could be made to see how perceptions of landscape and culture have shifted since 2019. I saw no issues in this paper and hope it is published soon.

Author Response

Thank you for your positive response to our manuscript and interest in our research. In an ongoing research project, partly based in the same case study area, we examine the implementation of ecosystem services with focus on the landscap´s cultural values.

The manuscript has been reviewed by a professional language reviewer before submission. If the editor and reviewers think I should send the manuscript one more time to the language reviewer, I'm happy to do so. But in that case, it will take another few weeks before the language review is complete.

Reviewer 2 Report

This is an excellent paper with only minor improvements being needed. The improvement that is needed is to contextualize this research within the broader endeavor sustainability in environmental planning/cultural landscapes because similar projects have been done in other parts of the world. Sources of consult are:

Cristina Herrero-Jáuregui and María Fe Schmitz in Cultural Landscapes Preservation and Social-Ecological Sustainability

Constanza Parra, Inger Birkeland, Katriina Siivonen, and Rob Burton in Cultural Sustainability and the Nature-Culture Interface

Amalia Leifeste and Barry L. Stiefel in Sustainable Heritage: Merging Environmental Conservation and Historic Preservation

Author Response

Thank you for your positive response to our manuscript and suggestion of literature which have improved the paper.

The introduction has been restructured and references are added both to the introduction and to the discussion.

The manuscript has been reviewed by a professional language reviewer before submission. If the editor and reviewers think I should send the manuscript one more time to the language reviewer, I'm happy to do so. But in that case, it will take another few weeks before the language review is complete.

Reviewer 3 Report

This is an interesting work on underdeveloped topics, and it is difficult to present precise conclusions.

The title is inaccurate and incomprehensible. Since this is a case study, this must be made clear by the title and should therefore be amended.

The introduction should be restructured. The way it is presented becomes confusing and does not meet the objectives of an introduction. It must be improved.

In terms of the construction of the article I would have liked to have seen a review of the literature (which also does not exist) associated with the construction of the conceptual scheme that does not exist.

The sample is too small to be able to prove anything and the authors present many results and conclusions that have no support or can be proven.

They should also have made similar views with other similar studies.

The displayed graphs should be centred and visually not work.

The bibliography must also be formatted.

Author Response

Thank you for your comments which I think have improved the paper. Changes have been made in relation to the time limit given for the minor revision.

The title has been changed.

The introduction has been restructured and revised and references are added to the introduction and to the discussion.

The conclusion has been restructured and revised. and changed to “Concluding remarks” in the discussion chapter. According to the instructions given by MPDI a conclusion chapter is not mandatory.

The bibliography has been formatted.

The graphs have been centred.

The manuscript has been reviewed by a professional language reviewer before submission. If the editor and reviewers think I should send the manuscript one more time to the language reviewer, I'm happy to do so. But in that case, it will take another few weeks before the language review is complete.

Reviewer 4 Report

The Authors of the article „The price of ignoring historic and cultural dimensions of landscapes - discussing sustainability in environmental planning and management” examines the role of historic and cultural landscape dimensions and its relation to the recent implementation of the ecosystem service framework in environmental practice. They used semi-structured interviews with officials at local and regional planning levels in a Swedish case study and showed that the historical landscape forms the basis for the environ- mental work. The topic addressed in the article is current and in line with global research trends.

The article is a very interesting study. The description of the research carried out, the results obtained, along with the discussion of the results and the final conclusions are not questionable.

My minor comments, which I hope will improve the quality of the article:

1. I miss a bit more detailed presentation of solutions, within the framework of the analyzed topic, from other countries. While the authors refer to international publications, they do not relate in more detail to specific studies conducted in other countries. I would suggest adding that in the Chapter 1 Introduction, or in the discussion of the results.

2. Please indicate and describe more clearly what innovative and novel things the authors propose, to what extent their research fills any gap in the current state of knowledge.

3. Personally, I am not in favor of separating a large number of very short subchapters. It would look better, for example, to remove the separation of chapter 2 (the content, of course, will remain at the end of the introduction), similarly with 3.2. By the way, please correct: the number 3.3 is missing?

4. For figure 1, please add scale and source

5. I'm not a language expert, but I would suggest analyzing the article linguistically (e.g., the final conclusions have different stylistic forms, I don't know if this is a translation issue or a stylistic error)

6. The article needs editorial corrections (e.g. bad form of references) bringing the article in line with MDPI editorial requirements. But this will certainly be brought to the attention of the Editor.

 Having taken into account the above minor comments, I recommend the article for publication.

Author Response

The introduction has been restructured and revised and references are added to the introduction and to the discussion.

The gap of knowledge is now presented in the Introduction and Discussion chapters.

The conclusion has been restructured and revised and changed to “Concluding remarks” in the discussion chapter. According to the instructions given by MPDI a conclusion chapter is not mandatory.

The manuscript has been restructured to avoid short subchapters.

Figure 1 has been changed.

The manuscript has been reviewed by a professional language reviewer before submission. If the editor and reviewers think I should send the manuscript one more time to the language reviewer, I'm happy to do so. But in that case, it will take another few weeks before the language review is complete.

The bibliography has been formatted.